# Massive palmitoylation-dependent endocytosis during reoxygenation of anoxic cardiac muscle

Mei-Jung Lin[1], Michael Fine[1], Jui-Yun Lu[2], Sandra L Hofmann[2], Gary Frazier[3], Donald W Hilgemann[1]*

[1]Department of Physiology, University of Texas Southwestern Medical Center, Dallas, United States; [2]Department of Internal Medicine and the Hamon Center for Therapeutic Oncology Research, University of Texas Southwestern Medical Center, Dallas, United States; [3]Department of Physics, University of Texas at Dallas, Richardson, United States

**Abstract** In fibroblasts, large Ca transients activate massive endocytosis (MEND) that involves membrane protein palmitoylation subsequent to mitochondrial permeability transition pore (PTP) openings. Here, we characterize this pathway in cardiac muscle. Myocytes with increased expression of the acyl transferase, DHHC5, have decreased Na/K pump activity. In DHHC5-deficient myocytes, Na/K pump activity and surface area/volume ratios are increased, the palmitoylated regulatory protein, phospholemman (PLM), and the cardiac Na/Ca exchanger (NCX1) show greater surface membrane localization, and MEND is inhibited in four protocols. Both electrical and optical methods demonstrate that PTP-dependent MEND occurs during reoxygenation of anoxic hearts. Post-anoxia MEND is ablated in DHHC5-deficient hearts, inhibited by cyclosporine A (CsA) and adenosine, promoted by staurosporine (STS), reduced in hearts lacking PLM, and correlates with impaired post-anoxia contractile function. Thus, the MEND pathway appears to be deleterious in severe oxidative stress but may constitutively contribute to cardiac sarcolemma turnover in dependence on metabolic stress.

*For correspondence: donald.
hilgemann@utsouthwestern.edu

**Competing interests:** The authors declare that no competing interests exist.

**Reviewing editor**: Richard Aldrich, The University of Texas at Austin, United States

## Introduction

Acute ischemic events are now a leading cause of death in the entire world (*World Health Organization, 2013*). Ironically, much of the damage from ischemic events can occur during reoxygenation of tissue when mitochondria begin to generate reactive oxygen species (ROS) at increased rates (*Jennings, 2013*). One of the hallmarks of reperfusion injury is a pronounced swelling of mitochondria, and the extensive opening of mitochondrial permeability transition pores (PTPs) that underlies this swelling (*Haworth and Hunter, 1979*) is a nearly decisive step on the path to cell demise (*Halestrap et al., 2004*). Associated with this progression, mitochondria release cytochrome c to the cytoplasm and thereby activate apoptosis programs that can definitively establish cell fate (*Gottlieb, 2011*).

Given that both outer and inner mitochondrial membranes become permeable during reperfusion injury, mitochondria with certainty release a large number of metabolites and small proteins. Therefore, the question arises whether, besides cytochrome c, other molecules released from mitochondria might serve signaling functions. Our analysis of Ca-activated endocytosis in BHK fibroblasts (*Hilgemann et al., 2013*) suggests that mitochondrial release of coenzyme A (CoA) may be important. This hypothesis arises from two facts. First, mitochondria accumulate CoA to high concentrations with respect to the cytoplasm (*Leonardi et al., 2005*). Second, many key cytoplasmic enzymes are modulated by binding long-chain acyl CoA (*Faergeman and Knudsen, 1997*), a CoA metabolite that will be

**eLife digest** Many people who survive a stroke or heart attack experience substantial tissue damage when the blood supply is restored. Much of this damage can be caused by the mitochondria inside the cells releasing a protein called cytochrome c that can cause cells to die in a process called apoptosis. The cytochrome c is released as the outer membrane of the mitochondria becomes permeable and pores called permeability transition pores open up in the inner membrane. Now Lin et al. explore if additional molecules released from the mitochondria might also initiate important cellular responses during the reoxygenation of oxygen-deprived tissue.

Lin and co-workers recently showed that the mitochondria of some cells can release a small enzyme cofactor, coenzyme A, which then promotes a cellular response called massive endocytosis. This process can cause up to 70% of the cell surface membrane to be absorbed into the interior of the cell in the form of membrane vesicles. Most forms of endocytosis involve a much smaller fraction of the cell membrane and employ a set of well-known endocytic proteins that are not involved in massive endocytosis. Now, Lin et al. investigate the role of massive endocytosis in cardiac muscle.

Electrical and optical measurements reveal that massive endocytosis occurs as cardiac cells that have been deprived of oxygen are reoxygenated. Lin et al. also find that an enzyme called DHHC5 must be present to allow endocytosis to take place during reoxygenation. DHHC5 is an enzyme that catalyzes a process called acylation – the transfer of acyl groups to proteins at the cell surface. Moreover, the deletion of DHHC5 has a beneficial impact on the performance of cardiac muscle after oxygen deprivation, which implies that molecules that inhibit protein acylation might protect the heart from damage during reoxygenation. Together, these results establish new pathological and physiological roles for the acylation, which is one of the most common biochemical modifications made to membrane proteins after they are synthesized.

generated immediately if CoA is released (*Idell-Wenger et al., 1978*). These include glycogen synthase (*Wititsuwannakul and Kim, 1977*), glucose-6-phosphate dehydrogenase (*Kawaguchi and Bloch, 1974*), and perhaps AMP kinase (*Faergeman and Knudsen, 1997*), as well as KATP potassium channels (*Shumilina et al., 2006*). Our study in BHK cells (*Hilgemann et al., 2013*) suggests that release of CoA from mitochondria may promote the palmitoylation of surface membrane proteins, subsequent to generation of acyl CoA. Further, our study suggests that extensive palmitoylation promotes internalization of membrane as a form of 'lipid raft'-dependent endocytosis (*Doherty and McMahon, 2009*).

Beyond the function of PTPs at the threshold of cell death, it is now established that transient mitochondrial PTP openings play important physiological functions by causing transient mitochondrial depolarizations that release matrix Ca to the cytoplasm accompanied by the generation of superoxide flashes (*Brenner and Moulin, 2012*; *Zhang et al., 2013*; *Zhou and O'Rourke, 2012*). In this light, a possibility emerges that cytoplasmic free CoA could physiologically play important second messenger functions in dependence on its release from mitochondria. On the one hand, mitochondrial depolarization will favor reverse CoA transport to the cytoplasm (*Tahiliani, 1991*), and on the other hand transient PTP openings may allow enough CoA to escape from the matrix space, down its more than 50-to-1 gradient, to significantly increase the low micromolar concentration of free cytoplasmic CoA (*Idell-Wenger et al., 1978*).

We describe here experiments that address these possibilities in cardiac myocytes and in intact cardiac tissue. First, we show that the function and abundance of cardiac Na/K pumps and Na/Ca exchangers is affected by the expression of the plasmalemma-directed acyl transferase, DHHC5, then we show that massive endocytic (MEND) responses in isolated myocytes depend on acyl CoA and DHHC5 activity, and that MEND responses occur when PTP openings occur during the reoxygenation of anoxic cardiac muscle. Finally, we show that two cardiac membrane proteins are indeed increasingly palmitoylated during reoxygentation, that internalized PLM (phospholemman) is more heavily palmitoylated than PLM on the cell surface, and that DHHC5 activity significantly impacts cardiac contractile recovery after anoxia.

## Results

In developing a tactic to characterize the MEND pathway in cardiac myocytes, we chose to focus on Na/K pumps. First, Na/K pumps are high density transmembrane protein complexes in the cardiac sarcolemma whose activity strongly modulates cardiac function (*Hilgemann, 2004*). As such, Na/K pumps likely represent a large fraction of membrane particles that have been visualized in freeze fracture studies and that cluster and decrease in number during anoxia/reoxygenation episodes (*Frank et al., 1980*). Second, it was described almost 40 years ago that the number of Na/K pumps, quantified as ouabain binding sites, decreases markedly during ischemia/reperfusion events (*Beller et al., 1976*). Recently, it has been shown that Na/K pumps are indeed internalized via endocytosis in a cardiac myocyte culture during 'simulated' ischemia/reperfusion (*Belliard et al., 2012*), and internalization still occurs during metabolic stress when a di-leucine motif of Na/K pump alpha subunits has been mutated to prevent pump involvement in conventional, clathrin-dependent trafficking (*Pierre et al., 2010*). Third, extensive internalization of Na/K pumps also occurs in response to metabolic stress in several other cell types (*Mandel et al., 1994*; *Bortner et al., 2001*; *Brown et al., 2001*; *Mann et al., 2001*). In alveolar epithelia the reversible internalization of Na/K pumps in hypoxia is initiated by ROS generation and the activation of protein kinase Cs (PKCs) (*Dada et al., 2003*), details that are reminiscent of our findings that PKCs and ROS promote MEND in BHK cells (*Hilgemann et al., 2013*). Fourth, the FXYD subunits of Na/K pumps can be dually palmitoylated and depalmitoylated at the surface membrane, and we have shown that overexpression of the major FXYD protein of cardiac muscle, phospholemman (PLM), promotes the occurrence of MEND in a PKC-dependent fashion in a TReX-293 cell line (*Hilgemann et al., 2013*).

As a second tactic, we developed a new electrical method to monitor surface membrane area as electrical cell capacitance ($C_m$) *on-line* in intact tissue over periods of hours. In results presented, we employ this non-invasive $C_m$ (NIC) recording technique with superfused right ventricular strips (see 'Materials and methods'). Briefly, NIC recording exploits the fact that myocyte sarcolemma constitutes the great majority of cellular surface membrane in the heart (*Banerjee et al., 2007*). Although non-myocytes can be as numerous as myocytes, all non-myocytes are much smaller than myocytes. Similar to skeletal muscle, therefore (*Cole, 1976*), rapid extracellular voltage oscillations (15 kHz/1 mV) can be used to monitor the composite sarcolemmal $C_m$. The use of this *on-line* method allowed us to rapidly test for the occurrence of MEND in intact cardiac tissue. As described subsequently, we have verified all findings from NIC recording with independent methods, specifically via an optical method in intact hearts to follow fluid phase uptake of fluorescent probes and via patch clamp of isolated myocytes. Additional results that validate NIC recording are provided in 'Material and methods'.

### Sarcolemmal significance of constitutive DHHC5 activity

*Figure 1* demonstrates that expression of the DHHC5 acyl transferase strongly influences Na/K pump activity in cultured cardiac myocytes, namely human fibroblast-derived cardiac myocytes (iCell Cardiomyocytes, [*Ma et al., 2011*]). These myocytes are transfected easily to overexpress and knockdown regulatory proteins, they highly express cardiac-specific proteins, such as PLM and the Na/Ca exchanger, NCX1, and ion transport activities are equivalent to those of adult myocytes (*Fine et al., 2013*). Measuring maximal Na/K pump currents exactly as described in our study (*Fine et al., 2013*), the first two bar graphs of *Figure 1* show that overexpression of DHHC5, together with green fluorescent GFP to identify transfected myocytes, decreases Na/K pump currents by 55%. This decrease becomes still somewhat larger with dual DHHC2/DHHC5 expression. Conversely, knockdown of DHHC5 by siRNA causes a 38% increase of Na/K pump currents, compared to myocytes transfected with scrambled siRNA ('siRNA control'), and dual DHHC5/DHHC2 knockdown causes a 90% increase. Although these changes may arise from either a change of pump activity or a change of pump density in the sarcolemma, data presented next indicate that changes of transporter localization play a large role.

We next describe myocytes and cardiac tissue from mice that are homozygous for a hypomorphic allele of DHHC5 (gene-trapped, GT) with cardiac DHHC5 expression decreased by >80% (*Li et al., 2011*). DHHC5-GT mice have significantly reduced growth rates. Between 4 and 5 weeks, the average weights of DHHC5-GT mice (14 ± 0.8 g) were 24% less than control litter mates (19 ± 0.9 g). Between 12 and 14 weeks, the average weights of DHHC5-GT mice (21 ± 0.2 g) were 13% less than control litter mates (24 ± 0.2 g). The dimensions of isolated myocytes from young DHHC5-GT animals were also

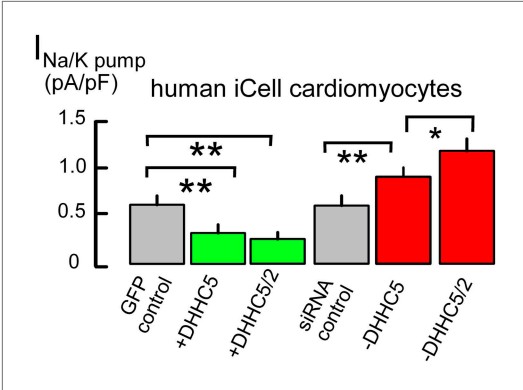

**Figure 1**. Na/K pump current densities are inversely dependent on DHHC5 expression in human iCell Cardiomyocytes. From left to right, Na/K pump current densities in iCell Cardiomyocytes transfected with GFP to identify transfected cells, DHHC5, DHHC5 and DHHC2, transfected with scrambled siRNA, siRNA for DHHC5, and siRNA for DHHC5 and DHHC2. Na/K pump currents are decreased by 55% and 61% with DHHC5 and DHHC5/DHHC2 overexpression, respectively. Current densities are increased by 38% with DHHC5 knockdown and by 90% with DHHC5/DHHC2 knockdown, n >6 for all results.

significantly smaller than those from matched WT animals. Therefore, we employed animals 12 to 14 weeks of age for the studies to be described.

We used patch clamp with square wave voltage pulses to measure the myocyte surface area as cell electrical capacitance ($C_m$) (*Lariccia et al., 2011*), assuming that 1 pF constitutes 100 $\mu m^2$ of membrane. Cell area (i.e., $C_m$) /volume ratios of myocytes were then determined as described in *Figure 2A*. The volume of each myocyte was estimated from a micrograph of the myocyte still resting on the surface of the chamber employed. From the micrograph, the two-dimensional area of the myocyte was determined in square microns using Image-J software (http://rsbweb.nih.gov/ij/). The volume was then estimated by assuming an average 5-to-1 ratio of myocyte width to thickness (volume ($\mu m^3$) = myocyte area$^2$/myocyte length/5) and converted to pl. As shown in bar graphs in *Figure 2B,C*, the average surface areas of DHHC5-GT myocytes were modestly but significantly increased in comparison to WT myocytes from matched animals. The average volumes of DHHC5-GT myocytes were insignificantly decreased in myocytes from 12 to 14 week old animals. As shown in *Figure 1D*, the average myocyte area/volume ratio, calculated in pF/pl, was highly significantly increased by 24%. *Figure 2E* shows that comparable results were obtained via NIC recordings in right ventricles from DHHC5-GT and litter mate mice. The capacitive NIC signal (i.e., sarcolemma area in a hemisphere below the 0.6 mm recording electrode) was increased significantly by 21%, on average, in DHHC5-GT ventricles (n = 5) vs WT ventricles (n = 5).

Next we show in *Figure 2F,G* that the average current density of Na/K pumps is significantly increased by 37% in DHHC5-GT myocytes, while a 17% increase of Na/Ca exchange (NCX1) current density was not significant. Finally, we tested directly whether pumps and exchangers are on average more retained in the sarcolemma vs internal membranes in DHHC5-deficient vs WT hearts. To do so, we employed a PEGylation assay described in 'Materials and methods' to determine the fractions of NCX1 and the regulatory Na/K pump subunit, PLM, resident in the cell surface. Both of these proteins are advantageous for amine PEGylation because they have extracellular N-termini (*Geering, 2006*; *Ren and Philipson, 2013*). Using a 5 kD NHS-PEGylation reagent to label the outer sarcolemma of intact, perfused hearts, *Figure 2H* shows typical Western blot density profiles from this assay for PLM and NCX1. Using a least squares Gaussian fit to determine relative protein densities, 34 ± 3% of PLM and 31 ± 2% of NCX1 bands were shifted 5 kD to higher molecular weights in hearts from WT animals. As shown in *Figure 3B*, the calculated surface fractions of both membrane proteins were significantly increased in DHHC5-GT hearts vs WT hearts, namely by 27 ± 5 and 23 ± 2%, respectively.

## DHHC5 deficiency inhibits MEND in four protocols in murine myocytes

In BHK cells, large MEND responses can be evoked by rapidly perfusing the cytoplasm of cells via the patch clamp pipette with solutions containing metabolites that promote mitochondria to open PTPs ('KSP solution', containing potassium [100 mM], succinate [5 mM], and inorganic phosphate [Pi, 1 mM]). As shown in *Figure 3A*, WT myocytes generate 33% MEND responses upon pipette perfusion of KSP solution with rough time constants of 2.5 min, and these responses are decreased significantly by 75% in DHHC5-GT myocytes. When the cytoplasmic solution employed in BHK cells contains myristoyl CoA (mCoA), application of $H_2O_2$ (hydrogen peroxide) (80 $\mu M$) for a few minutes primes cells to undergo MEND when the oxidative stress is removed. As shown in *Figure 3B*, transient $H_2O_2$ application (80 $\mu M$) also causes large MEND responses in cardiac myocytes (~35%) when mCoA (15 $\mu M$) is included in the pipette. These responses were also strongly reduced in DHHC5-GT myocytes (p<0.01).

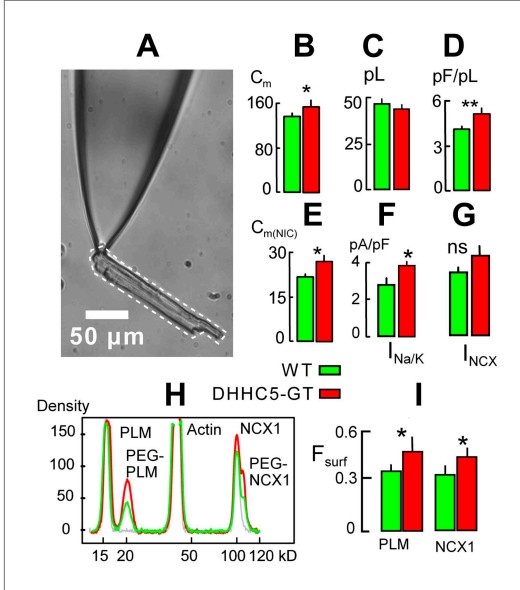

**Figure 2**. DHHC5-GT cardiac myocytes.
(**A**) Representative myocyte micrograph used to determine myocyte area/volume ratios. 25 × LWD lens. (**B**) Myocyte surface areas ($C_m$) are significantly increased in DHHC5-GT myocytes vs matched WT myocytes. (**C**) Myocyte volumes are not significantly different in DHHC5-GT myocytes from 4 to 6 week old mice. (**D**) Myocyte surface area/volume ratios are increased by 27% in DHHC5-GT myocytes. (**E**) Cardiac tissue $C_m$, monitored via NIC recording, is increased by 21% in right ventricular strips from DHHC5-GT mice (n = 5) vs control litter mates (n = 5). (**F**) Na/K pump current densities are increased by 32% in DHHC5-GT myocytes. (**G**) An average 17% increase of NCX1 current density is not significant. (**H**) Amine PEGylation assay to determine the surface membrane fractions of PLM and NCX1 in WT and DHHC5-GT hearts. Western blot density profiles are shown for three hearts, one control heart that was not PEGylated (gray), one WT heart (black) and one DHHC5-GT heart (red). PLM, actin and NCX1 are blotted, and the PEGylated PLM and NCX1 densities (i.e., protein resident in the cell surface) are shifted 5 kD from control densities. (**I**) Fractions of PLM and NCX1 that can be PEGylated, and therefore reside in the sarcolemma, are increased by 27 and 23%, respectively, in DHHC5-GT hearts. For both WT and DHHC5-GT hearts, n = 3. For all data from myocytes, n >40 using myocytes from three animals.

In BHK cells, activation of PKCs with diacylglycerol surrogates induces MEND responses that constitute 20–40% of the plasmalemma when mCoA is present in cytoplasmic solutions (*Hilgemann et al., 2013*). The equivalent experiments caused much slower responses in myocytes, as well as when phospholipase C-coupled GPCRs (G-protein coupled receptor) were activated (e.g., by adenosine or phenylephrine) in the presence of mCoA. *Figure 3C* illustrates MEND responses that occur when the alpha receptor agonist, phenylephrine (50 µM) (*Puceat et al., 1994*), is used in the presence of mCoA (20 µM). MEND did not occur unless both phenylephrine and mCoA were present (n = 10). While slower than responses to KSP solution and transient oxidative stress, sarcolemma area decreases on average by about 40% over 30 min in WTI myocytes, but only 14% in DHHC5-GT myocytes (p<0.05).

In murine myocytes, Ca influx can activate MEND responses that occur much more rapidly than those just described (*Lariccia et al., 2011*). When activated by reverse Na/Ca exchange, MEND stops abruptly when Ca influx is terminated (*Lariccia et al., 2011*), suggesting that Ca is acting via a rather direct mechanism that is likely different from MEND responses just described. Nevertheless, *Figure 4A* demonstrates that Ca-activated MEND is highly significantly reduced in DHHC5-GT myocytes vs WT myocytes. *Figure 4A* illustrates Ca-activated MEND in a WT myocyte, the top trace being $C_m$ (i.e., membrane area) and the bottom trace being membrane current. When Ca influx is activated, $C_m$ begins to decline after a 3 to 4 s delay and then declines further by 28% with a rough time constant of 8 s. As documented in *Figure 2G*, exchange currents were on average larger in DHHC5-GT myocytes. However, as shown in *Figure 4B*, MEND responses were decreased from 27% on average in WT myocytes to only 3% in DHHC5-GT myocytes. *Figure 4B* shows further that MEND responses in WT myocytes were decreased by 70% when myocytes were preincubated with the PTP/cyclophillin D-specific cyclosporine, NIM811 (N-methyl-4-isoleucine cyclosporine) (3 µM) (*Waldmeier et al., 2002*), for 1 hr. However, inclusion of a high CoA concentration (4 mM) in the cytoplasmic solution to *acutely* inhibit DHHC acyl-transferease activity (*Hilgemann et al., 2013*) did not significantly inhibit Ca-activated MEND in WT myocytes. We conclude therefore that palmitoylation reactions do not occur during the protocol of *Figure 4A*, although this form of MEND nevertheless depends on the palmitoylation state of the sarcolemma.

*Figure 4C* solidifies this interpretation using myocytes that overexpress Na/Ca exchangers (NCX1) by 10-fold. Both the time course of MEND and the decay of NCX1 current during Ca influx are accelerated nearly 10-fold (τ <1 s), and myocytes internalize more than 50% of their sarcolemma within

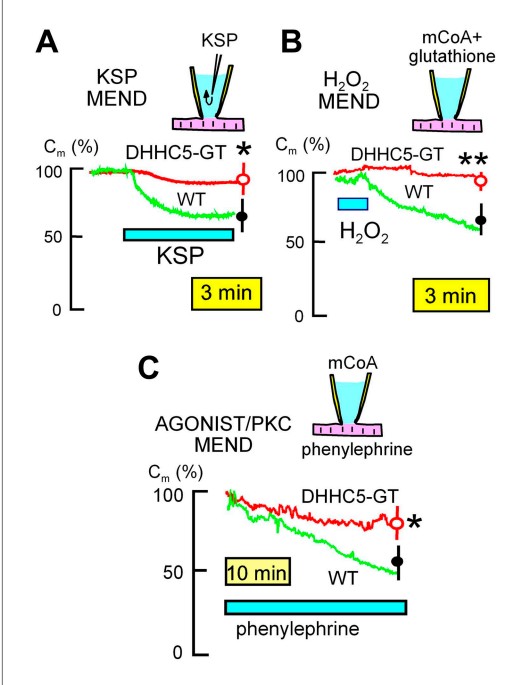

**Figure 3**. MEND evoked by three different stimuli is inhibited in DHHC5-deficient myocytes. (**A**) **KSP MEND**. Cytoplasmic application of KSP solution via the patch pipette causes MEND responses in WT myocytes that take place with rough time constants of 2 min and that amount to 24% of the sarcolemma, on average. The KSP MEND responses are decreased by 74% on average in myocytes from DHHC5-GT animals. (**B**) **$H_2O_2$ MEND**. The application and removal of $H_2O_2$ (80 μM) results in 38% MEND responses, on average, with MEND occurring only after the oxidative stress is relieved. These MEND responses require that the cytoplasmic solution contains acyl CoA (mCoA, 15 μM). They are more reliable when glutathione (3 mM) is included in the cytoplasmic solution, presumably because final steps of endocytosis require a reducing environment. (**C**) **GPCR MEND**. MEND occurs slowly over 30 min in the presence of phenylephrine (30 μM) when the cytoplasm contains mCoA (20 μM). These endocytic responses, while slow, amount on average to >40% of the sarcolemma, and they are inhibited by 67% in myocytes from DHHC5-GT animals.

1–2 s. Using the myosin II inhibitor blebbistatin (*Farman et al., 2008*) (10 μM in all solutions) to inhibit contraction, myocytes did not contract or show obvious morphological changes during these profound responses. As shown in *Figure 4D*, neither pretreatment with NIM811 for 1 hr (3 μM) nor inclusion of a high CoA concentration (4 mM) in the cytoplasmic solution significantly affected these responses. Clearly, Ca can promote MEND in myocytes by a mechanism that is promoted by, but does not require, membrane protein palmitoylation.

## MEND occurs during reoxygenation of anoxic cardiac muscle

Reperfusion of ischemic cardiac tissue is one of the best defined circumstances in which PTP openings occur profusely and correlate with a significant amount of reoxygenation injury after ischemic or anoxic episodes (*Halestrap, 2009*). Importantly, much of this damage can be prevented by chemical preconditioning of cardiac tissue with hormones, such as adenosine, that activate PKCε and likely inhibit PTP openings (*Baines et al., 2003*). Given evidence that significant endocytosis can occur in cardiac ischemia/reperfusion via unconventional mechanisms (*Pierre et al., 2010*), we tested next whether MEND indeed occurs during reoxygenation of anoxic cardiac tissue.

*Figure 5A* illustrates NIC recording in rapidly superfused right ventricular strips (0.5–0.7 mm thick; 36°C). Using 15 kHz sinusoidal voltage oscillations, the conductance signal component ($I_G$) decreases and the capacitive signal component ($I_{CAP}$) increases as the perturbing electrode (a 0.6 mm Ag/AgCl pellet, recessed 0.4 mm in a glass capillary) is brought close to and touches the muscle surface. The conductance signal ($I_{GX}$) then reflects conductance of the extracellular space, which constitutes about 30% of the heart volume. Capacitive signals reflect the relative amount of surface membrane in a hemisphere of tissue below the electrode. The capacitive signal showed little change over 30 min after oxygen-saturated solution was switched to oxygen-free, glucose-free solution with 5 mM deoxyglucose to promote nucleotide depletion. However, when oxygen-saturated, glucose-containing solution was reintroduced, NIC signals decreased over 25 min by 21% on average.

Composite results from similar recordings, shown in *Figure 5B*, support the interpretation that MEND is occurring in intact cardiac muscle by mechanisms that are related to those described in BHK cells (*Hilgemann et al., 2013*). First, the cardiac MEND responses were strongly decreased when muscles were taken from hearts that had been FA (fatty acid)-depleted by 15 min arterial perfusion of FA-free albumin (60 μM) prior to the experiments. Second, a low concentration of staurosporine (STS; 0.1 μM), which promotes MEND responses in BHK cells (*Hilgemann et al., 2013*), increased

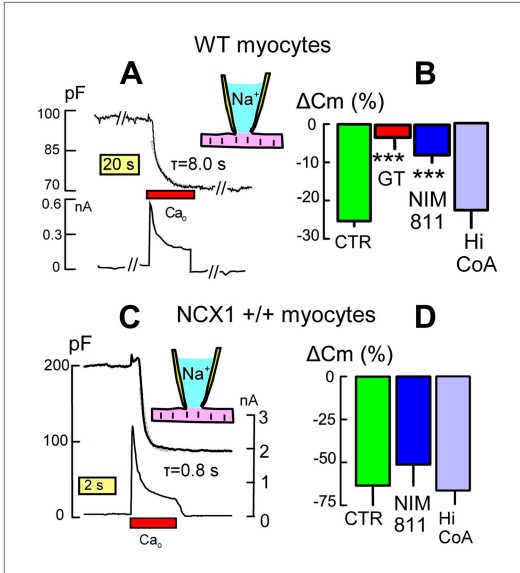

**Figure 4**. DHHC5 facilitates Ca-activated MEND in cardiac myocytes. (**A**) Typical endocytic response caused by Ca influx (i.e., reverse Na/Ca exchange current) in a WT myocyte. Top record, $C_m$ (i.e., membrane area); bottom record, membrane current. MEND occurs during Ca influx with a time constant of ~8 s. (**B**) Ca-activated MEND in WT myocytes amounts to 27% of the sarcolemma on average. MEND is reduced by >80% in myocytes from DHHC5-GT animals and reduced 70% by pretreatment of cells with NIM811 (3 µM) for 1 hr. However, MEND is not reduced by inclusion of a high CoA concentration (4 mM) to acutely block acyl transferace activity. (**C**) Typical MEND response in a cardiac myocyte over-expressing NCX1 by 10-fold. Top trace, $C_m$ (i.e., membrane area); bottom trace, membrane current. In these myocytes with four to sixfold larger Na/Ca exchange currents, Ca influx causes MEND that amounts to more than 50% of the sarcolemma in less than 2 s. (**D**) Large Ca transients overcome the dependence of MEND on palmitoylation. Ca-activated MEND in NCX1-overexpressing myocytes is unaffected by 1 hr treatment with NIM811 (3 µM) or by a high cytoplasmic CoA concentration (4 mM) to acutely inhibit palmitoylation.

sarcolemma loss to 35%. Third, cyclosporine A (CsA; 1 µM) strongly decreased sarcolemma loss. Fourth, the preconditioning hormone, adenosine (0.1 mM) (*Ytrehus et al., 1994*), decreased sarco-lemma loss by 52%. And fifth, MEND was nearly ablated in muscles from DHHC5-GT animals.

As described in the accompanying article (*Hilgemann et al., 2013*), MEND appears to be 'cargo-dependent', being promoted by overex-pression of the palmitoylated Na/K subunit, PLM (*Tulloch et al., 2011*), in T-Rex293 cells. Therefore, we examined whether the occurrence of MEND would be altered in cardiac muscle from PLM-deficient mice. As shown in the last bar graph of *Figure 5B*, reoxygenation-induced MEND was reduced by 58% in right ventricles from PLM knockout mice (*Bell et al., 2008*).

As an independent verification of the occur-rence of MEND, we describe in *Figure 6* the fluid-phase uptake of FITC-labeled dextran (4000 MW) by myocytes in arterially perfused hearts. Hearts were perfused retrograde with either FA free- or 2:1 myristate-loaded albumin (60 µM) during the same anoxia-reoxygenation protocol. During reoxygenation, hearts were perfused with FITC-dextran (0.5 mM) for 25 min followed by washout at room temperature for 10 min. Subsequently, epicardial myocytes were imaged several cell layers below the outer cardiac surface; the outer most myocytes were excluded because they are atypical, being enriched in cardiac stem cells (*Schlueter and Brand, 2012*). Hearts perfused with FA—free albumin reveal diffuse staining of a limited fraction of myocytes. This reflects non-specific dextran uptake as a result of reperfusion stress. Hearts perfused with FA reveal, in addition to diffuse staining of a fraction of myocytes, bright punctate staining that is indicative of dex-tran uptake into large endosomes and vacuoles. Punctate staining becomes very profuse in hearts perfused with STS (0.1 µM) and is absent when CsA (3 µM) is perfused with albumin and FA. Statistical analysis, performed as described in Methods, is provided in *Figure 6B*, together with the additional result that dextran uptake was negligible in the same protocol in ventricles from DHHC5-GT mice.

## Palmitoylation is activated during reoxygenation-induced cardiac MEND

To test whether palmitoylation of membrane-associated proteins is indeed enhanced during reoxy-genation of anoxic cardiac tissue, we employed a resin assisted capture assay for acylated proteins (acyl-RAC) (*Forrester et al., 2010*) to determine the palmitoylation status of proteins that might be important in MEND. In addition to PLM, we examined flotillin-2 because flotillins may be involved in the ordering of proteins into *Lo* domains (*Langhorst et al., 2005*). In this assay, free cysteines are initially blocked, followed by deacylation of palmitoylated residues with hydroxylamine ($NH_2OH$), cysteine-specific pull-down, and quantification of captured, deacylated proteins via Western blotting. The presence of a single acylation site is 'positive' and multiple vs single acylations are not distinguished.

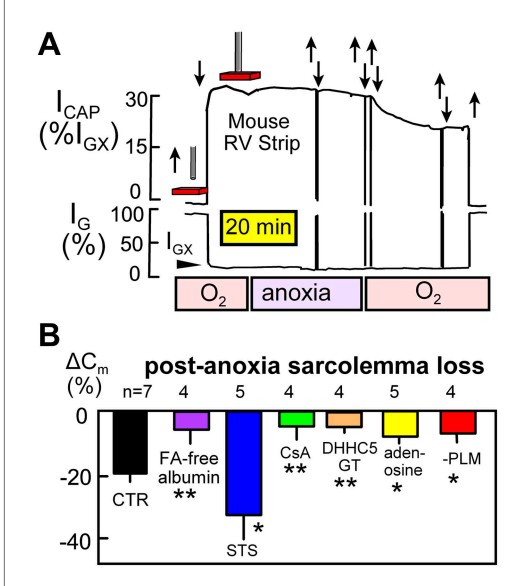

**Figure 5**. Electrical recordings of MEND during reoxygenation of anoxic cardiac muscle. (**A**) Noninvasive $C_m$ (NIC) recording in superfused right ventricular strips. The capacitive signal ($I_{CAP}$), reflecting sarcolemmal area in a hemisphere of tissue beneath the electrode, is stable during 30 min periods of anoxia, but the NIC signal decreases on average by 21% over 25 min upon reoxygenation of the tissue. (**B**) Loss of sarcolemma during reoxygenation for 25 min after a 30 min period of anoxia. The first bar graph quantifies the composite response for control (WT) muscles. Subsequent bars document from left to right that MEND is strongly inhibited by perfusing hearts for 15 min with FA free albumin (60 µM) before isolating muscle strips, is strongly enhanced by a low concentration (0.1 µM) of staurosporine, is potently inhibited by cyclosporine (CyS) (1 µM), is strongly reduced in muscles from DHHC5-deficient mice, is strongly reduced by the 'preconditioning' hormone, adenosine (100 µM), and is strongly reduced in cardiac muscle lacking PLM.

Proteins that are stably palmitoylated, such as caveolins (**Parat and Fox, 2001**), can be used as loading controls in the assay (**Tulloch et al., 2011**). Usually, we analyzed caveolin-3 (CAV3) content after stripping secondary antibodies employed in initial blots, but results were very similar when analyzed in relation to Western blot loading controls of initial lysates.

*Figure 7A* illustrates Western blots for a pair of hearts. One was subjected to the anoxia/reoxygenation protocol of *Figure 5A*, and the second heart was perfused with oxygen-containing perfusate for an equivalent time (1 hr at 1.5 ml/min; 37°C). For PLM blotting, we employed an N-terminal (extracellular) antibody (see 'Materials and methods') to avoid effects of cytoplasmic (C-terminal) phosphorylation or palmitoylation on antibody binding. As shown in *Figure 7B*, anoxia/reoxygenation caused a 71% increase of palmitoylated PLM (n = 5; p<0.01) and a 77% increase of palmitoylated flotillin-2 (n = 5; p<0.01) relative to caveolin-3. Consistent with the idea that DHHC5 constitutively affects palmitoylation of these proteins, *Figure 7C* shows that palmitoylated PLM and flotillin-2 were decreased by 27% and by 51%, respectively, in DHHC5-GT hearts that were perfused briefly to remove blood and immediately frozen.

We next combined the acyl-RAC assay with the amine PEGylation assay, illustrated in *Figure 2H*, to determine the fraction of PLM at the cell surface and then additionally determine the fraction of PLM that is palmitoylated at the surface membrane vs internal membrane pools. Using the PEGylation protocol described in 'Materials and methods', *Figure 7D* shows the PEGylation and acyl-RAC results for four hearts, two of which were PEGylated and two of which were not. The blots shown include a loading control and the acyl-RAC result for each heart with loading amounts adjusted to generate similar densities on the acyl-RAC blots. Densities of the loading controls from PEGylated hearts (second lanes from the left and from the right) indicate that 40 and 46% of PLM was PEGylated, respectively, and is therefore sarcolemmal. In the acyl-RAC pull-downs from the same hearts, however, only 17 and 23% of PLM is PEGylated. Therewith, a key tenet of our working hypothesis is verified. Although palmitoylation can target proteins to the surface membrane (**Greaves et al., 2009**), enhanced palmitoylation may favor the removal of palmitoylated proteins from the surface membrane.

## MEND impacts post-anoxia cardiac function

As described in *Figure 5B*, MEND is decreased in intact cardiac tissue by treatments that protect the heart from reperfusion damage, namely CsA and adenosine (**Halestrap et al., 2004**). Furthermore, we find that MEND is promoted by an agent that prevents ischemic preconditioning and promotes PTP openings, STS (**Ytrehus et al., 1994**). These results therefore suggest that the occurrence of MEND might negatively impact the recovery of the heart from anoxia-reoxygenation episodes. To test if this is indeed the case, we analyzed contractile function of right ventricular strips paced at 0.25 Hz and subjected to 30 min of anoxia in the absence of glucose, followed by reoxygenation and

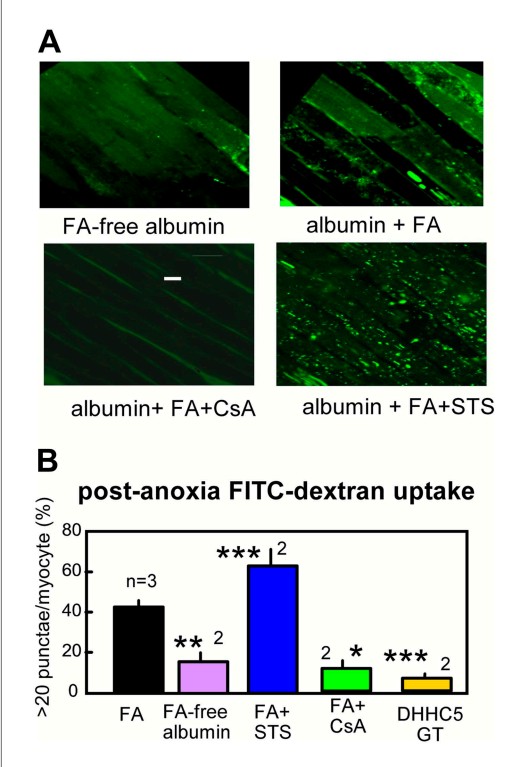

**Figure 6**. Optical recordings of MEND during reoxygenation of anoxic cardiac muscle. (**A**) Micrographs of FITC-dextran uptake in arterially perfused mouse hearts after 30 min of anoxia, followed by 25 min reoxygenation with perfusion of FITC-dextran (0.5 mM) and a 10 min washout at 25°C. Confocal images are from myocytes that are a few cell layers below the outer left ventricular cardiac surface. Scale bar, 30 µm. (**B**) Composite results for hearts with control perfusate (i.e., 0.1 mM albumin with 0.1 mM FA), with perfusate containing 0.1 mM FA-free albumin, with control perfusate containing cyclosporine (CsA, 2 µM), with control perfusate containing staurosporine (STS, 0.1 µM), and DHHC5-GT hearts with normal perfusate.

administration of glucose (15 mM). As shown in the examples in *Figure 8A* and in the composite data in *Figure 8B*, contraction became negligible within 8 min of anoxia without glucose. During reoxygenation for 25 min, contractile function of WT ventricles recovered only little, 12% on average. However, contractile function of right ventricles from DHHC5-GT animals recovered substantially, 60% on average, equivalent to some of the most effective 'preconditioning' protocols (*Halestrap et al., 2004*). Thus, as already suggested by others (*Belliard et al., 2012*), internalization of sarcolemma may contribute to the acute failure of cardiac function during ischemia/reperfusion episodes.

## Discussion

This article lends support to a working hypothesis that mitochondrial PTP openings can initiate surface membrane remodeling by promoting palmitoylation-dependent endocytosis (*Hilgemann et al., 2013*). We have demonstrated here that this form of endocytosis occurs extensively during the reoxygenation of anoxic cardiac tissue and therefore is relevant to the cellular consequences of ischemia/reperfusion events. In addition, DHHC5-dependent palmitoylation constitutively regulates the protein content of the cardiac sarcolemma and affects how cardiac myocytes respond to cell signals, such as oxidative stress.

### The constitutive function of DHHC5 in cardiac myocytes

From all molecular players in cardiac excitation-contraction coupling, small changes of Na/K pump activity cause the largest changes of myocyte Ca transients and therefore cardiac contractility (*Hilgemann, 2004*). In this light, Na/K pump activity changes caused by modifying DHHC5 and DHHC2 expression (*Figures 1 and 2*) appear relevant to the physiological, long-term control of basal cardiac contractility. It is certain that palmitoylation of the regulatory pump subunit, PLM, can cause pump inhibition (*Tulloch et al., 2011*). We have shown here, however, that the surface membrane fraction of PLM is increased by approximately the same extent as Na/K pump activity (*Figure 2*) in DHHC5-deficient myocytes. Furthermore, myocyte surface area/volume ratios are significantly increased. Analysis of ion channel and excitation-contraction coupling function are now required to define more clearly the roles of DHHC5 and DHHC2 in myocytes. However, the present results strongly suggest that DHHC5 activity modulates Na/K pump densities in cardiac myocytes by modifying pump internalization rates. In this connection, the fate of membrane internalized by DHHC5-dependent endocytosis remains to be determined. One possibility is that the pathway delivers surface membrane to autophagosomes (*Moreau and Rubinsztein, 2012*). That MEND responses are reduced in PLM-deficient cardiac tissue (*Figure 5B*) correlates well with the fact that MEND is enhanced when PLM is overexpressed in a model cell (*Hilgemann et al., 2013*). Therewith, a significant mechanistic involvement of Na/K pumps in the progression of MEND has been established. Clearly, proteomic analyses of palmitoylation by mass spectroscopy will be essential to evaluate the overall cellular scope of the MEND response.

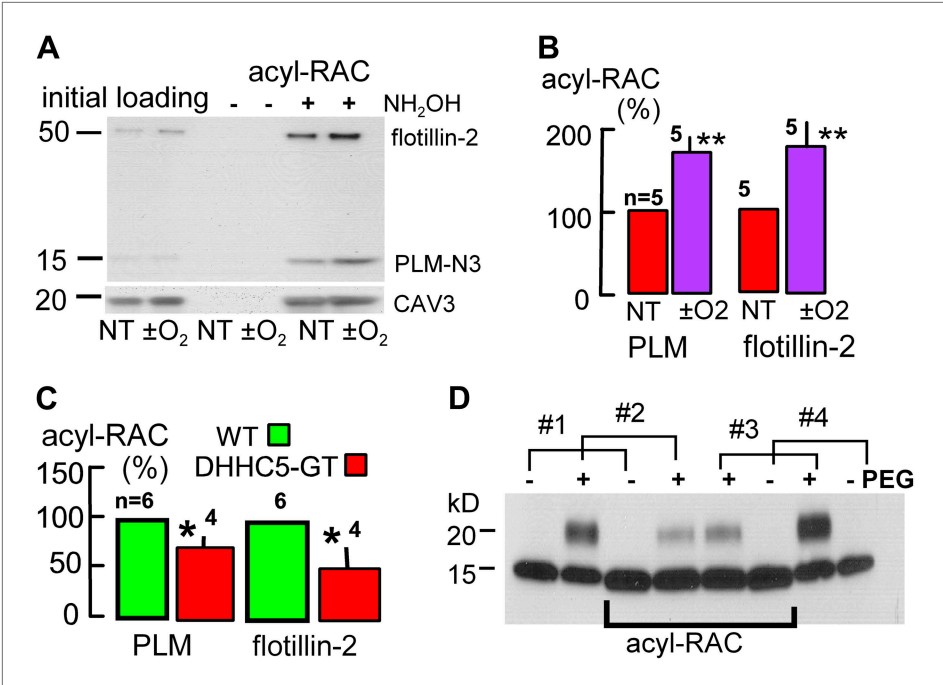

**Figure 7**. Palmitoylation is increased during reoxygenation-induced cardiac MEND. Palmitoylation of PLM and flottilin-2 during reoxygenation of anoxic cardiac tissue. (**A**) Experiments were performed in a pairwise fashion. One heart was quick-frozen after perfusion with oxygen for 45 min (not treated, NT) and one after being subjected to the anoxia/reoxygenation protocol, followed by biochemical analysis of palmitoylation. Left lanes show loading controls from initial lysates; right four lanes show samples after deacylation and precipitation with cysteine-reactive beads. No protein is detected without deacylation. Caveolin-3 was blotted on the same gel after stripping secondary antibodies. (**B**) Five paired data sets from experiments as in 'A', with palmitoylation calculated relative to palmitoylated caveolin-3 densities. (**C**) Constitutive palmitoylation of PLM and flotillin-2 in DHHC5-GT hearts, not subjected to anoxia/reoxygenation, is decreased 27 and 51%, respectively. Palmitoylation is not clearly changed in hearts subjected to 30 min anoxia without reoxygenation ('$-O_2$'). (**D**) Combined measurements of the surface membrane fraction and the palmitoylated fraction of PLM. Hearts #1 and #4 were not PEGylated; hearts #2 and #3 were PEGylated with PEGylated PLM fractions amounting to 40 and 46%, respectively, in the initial lysate. In the acyl-RAC pull-down of palmitoylated PLM, the surface membrane (PEGylated) fractions of PLM amount to only 17 and 23% in hearts #2 and #3, respectively. Thus, surface membrane PLM is substantially less palmitoylated than internalized PLM.

## Comparison of MEND in cardiac myocytes vs BHK fibroblasts

It is certain that mechanisms besides palmitoylation are sufficient to cause MEND in BHK cells. For example, extracellular lysolipids and sphingomyelinase C activities can cause MEND responses that are equivalent in magnitude to Ca-activated MEND (*Lariccia et al., 2011*). In our experience, these different mechanisms act synergistically, and the generation of submicoscopic membrane phase transitions appears to be the convergent effect that precedes endocytosis (*Hilgemann and Fine, 2011*). In the present study as well, it is likely that some MEND responses reflect synergistic effects of multiple mechanisms. As shown in *Figures 3 and 4*, the occurrence of MEND in four different protocols is strongly inhibited in DHHC5-deficient myocytes and therefore depends on palmitoylation. MEND occurs over one to several minutes in response to PTP-promoting metabolites, over several minutes in response to transient $H_2O_2$-induced oxidative stress, over 30 min in response to a GPCR agonist, phenylephrine, which activates GPCRs and subsequently PKCs, and over a few seconds when Ca transients are activated. In contrast to the delayed MEND responses that occur in BHK cells (*Hilgemann et al., 2013*), Ca transients cause immediate MEND responses in myocytes. These responses, which may represent myocyte wounding responses, are blunted in DHHC5-deficient myocytes when they are activated by moderate Ca transients. However, the explosive responses that

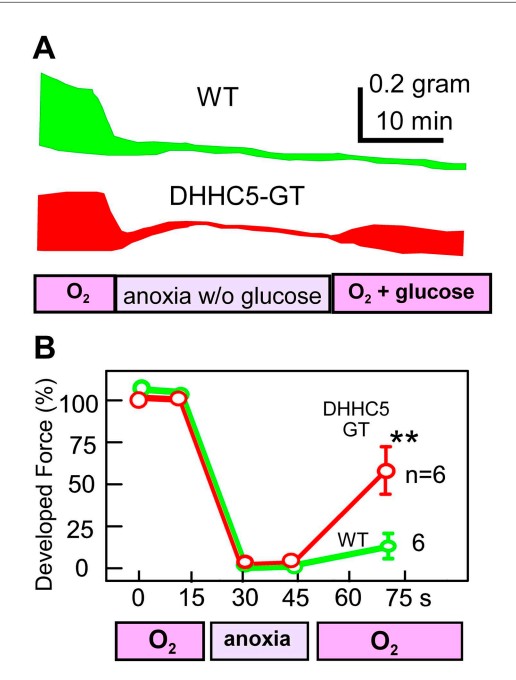

**Figure 8**. MEND correlates with impaired functional recovery of cardiac muscle from anoxia. Contractile function of mouse right ventricular strips, paced at 0.25 Hz, during anoxia for 30 min without glucose, followed by reoxygenation with glucose (15 mM) for 25 min. (**A**) Representative force envelopes of right ventricular strips. (**B**) Composite results for right ventricular strips from WT and DHHC5-GT mice. For both sets, n = 6.

occur when Ca transients are large (*Figure 4*) are unaffected by acute inhibition of DHHC5 activity. Thus, the palmitoylation state of myocytes promotes Ca-activated MEND, but Ca can cause MEND by a mechanism unrelated to DHHC activity. In contrast, myocyte MEND responses to transient oxidative stress and to phenylephrine require an immediate presence of cytoplasmic acyl Co A and therefore are likely to involve palmitoylation as they develop. In the case of MEND induced by KSP solution, we cannot discount at this time that mitochondrial Ca release is playing some role.

## MEND in intact cardiac muscle

In cardiac muscle more than 95% of total CoA metabolites reside in the mitochondrial matrix (*Idell-Wenger et al., 1978*), and the cytoplasmic free CoA concentration is about 7 µM (*Idell-Wenger et al., 1978*). Given that mitochondria make up about 30% of myocyte volume (*Zhou et al., 2011*), the physical basis for our working hypothesis is well established in myocytes. *Figures 5 and 6* document by independent methods that sarcolemma in intact hearts is significantly internalized upon reoxygenation after an anoxic episode. Internalization is dependent on fatty acids and DHHC5 activity. It is prevented by agents that precondition hearts against reoxygenation damage, namely adenosine and CsA, and it is promoted by low concentrations of staurosporine that promote reoxygenation damage (*Schneider, 2005*). The results together suggest that MEND responses occurring during reoxygenation may be relevant to human ischemia/reperfusion events.

As expected if MEND contributes significantly to myocyte damage during reoxygenation, contractile function of ventricular muscle is preserved by 60% in DHHC5-GT right ventricles after 30 min anoxia without glucose and 25 min reoxygenation with glucose (*Figure 8*). This degree of functional preservation is as good as standard 'preconditioning' protocols (*Cave et al., 1994*; *Schneider et al., 2001*). Whether the occurrence of MEND is in fact detrimental to functional recovery is uncertain. Loss of substantial amounts of sarcolemma may certainly impact cardiac excitation-contraction coupling. From a different perspective, MEND itself may not be the culprit. MEND appears to rely on the development of submicroscopic membrane phase transitions. Independent of endocytosis, therefore, the occurrence of membrane phase transitions may promote membrane defects and membrane shedding that mediates release of cytoplasmic enzymes and metabolites to the extracellular space. The important conclusion for reperfusion injury is that effective inhibition of palmitoylation may significantly inhibit this part of reperfusion damage, and these same considerations apply to the function of vascular cells and fibroblasts during reperfusion.

In conclusion, this article provides evidence that a signaling pathway leading from mitochondrial PTP openings to DHHC5 acyl transferases and endocytosis (*Hilgemann et al., 2013*) is activated during the reoxygenation of anoxic cardiac tissue. Multiple lines of evidence support that idea that endocytosis during reoxygentation requires increased membrane protein palmitoylation, and inhibition of the pathway appears to protect muscle from reoxygenation injury. The phenotypes of myocytes that are deficient in DHHC5 acyl activity suggest that the pathway contributes significantly to constitutive sarcolemma turnover and will mediate enhanced sarcolemma turnover in response to transient metabolic stress.

## Materials and methods

### Electrical methods and myocytes

Patch clamp (*Yaradanakul et al., 2008*) and myocyte preparation were as described (*Lariccia et al., 2011*). Software employed is available on-line (https://sites.google.com/site/capmeter/). The UT Southwestern Medical Center Animal Care and Use Committee approved all animal studies. Effects of voltage pulses (0.1–0.5 kHz) to determine $C_m$ are removed digitally from current records. For myocyte patch clamp and pipette perfusion experiments, highly polished pipette tips with diameters of >6 µm were employed. iCell myocytes (*Ma et al., 2011*) were obtained from CellularDynamics (Madison, WI) and cultured according to CellularDynamics instructions.

### NIC recording

NIC recording was performed using a conventional patch clamp circuit and phase-lock amplifier. Right ventricular strips were dissected at 10°C in a relaxing solution that contained 20 mM $MgCl_2$ to prevent electrical activity and contraction. Muscles were mounted horizontally in a chamber maintained at 37°C and superfused with physiological salt solution, described below, at a velocity of 1–2 cm/s. The capacitive component of NIC signals increased with increasing oscillation frequency up to a maximum at ~23 kHz. The optimal frequency for recording was therefore about 15 kHz. The phase angle was adjusted by up to 2° to insure that the capacitive signal was zero upon touching the probe to a soft rubber septum in the recording chamber. As described in *Figure 9*, capacitive signals in NIC recording can be strongly decreased by extracting lipids from right ventricular strips and are very insensitive to sarcolemmal conductance changes of very large magnitudes. One major experimental requirement for reliable NIC recording is that the probe remains mechanically stable with respect to the tissue and that conductance of the tissue (i.e., of the extracellular space) does not change. Conductance signal changes were negligible during NIC recordings presented in *Figure 5*.

### Solutions

Standard MEND Solutions minimize all currents other than NCX1 current. Extracellular solution contained in mM: 120 n-methyl-d-glucamine (NMG), 4 $MgCl_2$ ± 2 $CaCl_2$, 0.5 EGTA, 20 TEA-OH, 10 HEPES, pH 7.0 with aspartate. Cytoplasmic solution contained in mM: 75 NMG, 20 TEA-OH, 15 HEPES, 40 NaOH, 0.5 $MgCl_2$, 0.8 EGTA, 0.25 $CaCl_2$, 1 Pi set to pH 7.0 with aspartate. Unless stated otherwise, 8 mM MgATP, 2 mM TrisATP, and 0.2 mM GTP were employed in cytoplasmic solutions with a free Mg of 0.5 mM. KSP cytoplasmic solution contained 110 KOH, 40 NaOH, 10 histidine, 1.0 EGTA, 0.2 $CaCl_2$, nucleotides as just given, and pH 7.0 with aspartate. Bath solution in NIC recording and FITC-dextan uptake experiments contained in mM: 120 NaCl, 5 KCl, 0.5 $NaHPO_4$, 0.5 $MgCl_2$, 1.5 $CaCl_2$, 15 histidine, and 15 glucose. During anoxia, 5 mM deoxyglucose was substituted for glucose, 35 mM KCl was added to stop spontaneous activity, 50 µM FA-free albumin with 50 µM myristate was added, and solution was degassed by stirring under vacuum.

### Reagents and chemicals

Unless specified otherwise, reagents were from Sigma-Aldrich (St. Louis, MO). N-terminal ('N3') PLM antibody, a gift of Dr Will Fuller (Dundee), was raised in rabbit against the peptide, EAPQEPDPFTYDYHT, coupled to KLH by Moravian Biotechnology (Brno, Czech Republic) and affinity purified against the same peptide coupled to NHS-Sepharose.

### Statistics

Unless stated otherwise, error bars represent standard error of six and usually eight or more observations. Significance was accessed by Students $t$ test or, in rare cases of unequal variance, by the Mann-Whitney Rank Sum test. In figures, '*' denotes $p < 0.05$, '**' denotes $p < 0.01$, and '***' denotes $p < 0.001$.

### Acyl-RAC assays from perfused murine hearts

After completion of experimental protocols, hearts were perfused for 10 min with 10 mM NEM, flash-frozen in liquid nitrogen-cooled aluminum clamps, and stored at −80°C. Tissue was powdered and vortexed into lysis buffer (50 mM NaCl, 50 mM Hepes, 0.1% SDS, 1% NP-40, pH 7.4) with protease inhibitor cocktail (Roche), placed on ice for 15 min, and centrifuged at 16,000 × $g$ for 10 min. Supernatants were then collected and protein determined (Micro BCA kit, Pierce). Acyl-RAC assays were performed as described (*Forrester et al., 2010*) with modifications as follows. Duplicate sets of

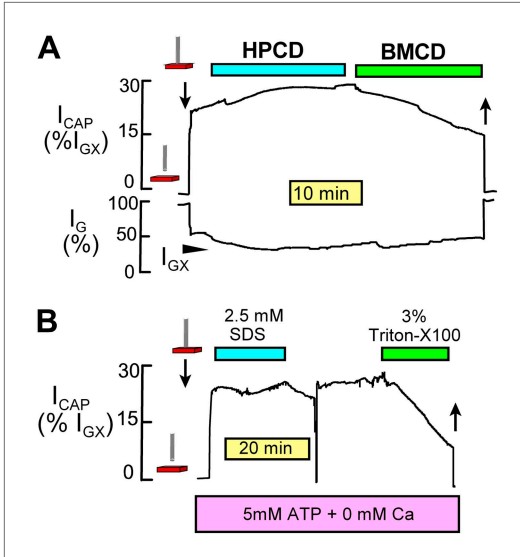

**Figure 9**. NIC recording monitors sarcolemmal area and is insensitive to sarcolemmal conductance changes. (**A**) As described previously (**Lariccia et al., 2011**), (2-hydroxypropyl)-β-cyclodextrin (HPCD) extracts cholesterol from fibroblasts without causing $C_m$ changes. However, methyl-β-cylodextrin (BMCD) causes large decreases of $C_m$ that presumably reflect extraction of both phospholipids and cholesterol. Similarly, NIC signals from right ventricular strips are unaffected by application of HPCD (8 mM) but are strongly decreased by application of BMCD (8 mM). (**B**) Muscle incubated with ATP and 0 Ca with 1 mM EGTA to avoid contraction when the sarcolemma develops ruptures. A rather high concentration of sodium dodecylsulfate (SDS, 2.5 mM) does not significantly decrease NIC signals. Although this concentration of SDS generates sarcolemmal leaks with great certainty, it does not effectively extract the sarcolemma from the intact muscle preparation. By contrast, Triton X100 at 3% very effectively decreases NIC signals from the same muscle, as expected for an effective membrane extraction. These results verify that NIC signals are not affected by increases of sarcolemmal conductance until cellular conductance becomes significant with respect to that of the extracellular space.

protein lysate were diluted to 2 mg/ml in 250 µl blocking buffer (100 mM Hepes, 1 mM EDTA, 2.5% SDS, 0.2% methyl methanethiosulfonate [MMTS], pH 7.5) and incubated at 40°C for 20 min with constant shaking. MMTS was removed by acetone precipitation at −30°C. Pellets were extensively washed with 70% ice-cold acetone and resuspended in 240 µl binding buffer (100 mM Hepes, 1 mM EDTA, 1% SDS, pH 7.5). The duplicate samples were separated, one was treated with 100 µl of thiopropyl Sepharose beads (GE Life Sciences) and 250 mM hydroxylamine, (HA) pH 7.5. The other was treated with 250 mM NaCl as negative control. Approximately 40 µl of each sample was retained as 'total input'. Samples were rotated at 23°C for 2.5 hr, beads were pelleted, and supernatants were retained as 'unbound' sample. Beads were extensively washed in binding buffer and acylated protein was eluted at 23°C in 50 µl binding buffer with 100 mM DTT for 20 min. Supernatant was removed, mixed with Laemmli loading buffer, heated at 37°C for 30 min, and separated on SDS-PAGE.

## PEGylation assay to determine the surface membrane fraction of PLM and NCX1

Hearts were perfused for 2 min to remove blood, followed by perfusion with 10 mM PEG succinimidyl ester (MW 5000; Nanocs, PG1-SC-5k) dissolved in 130 mM NaHCO$_3$ with 4 mM MgCl$_2$ at pH 8.7 for 15 min at 22°C and further incubation of hearts in the same solution at 5°C without perfusion. Thereafter, NHS was quenched by renewed perfusion of the bicarbonate perfusate diluted 1:3 with isotonic (300 mM) glycine solution for 15 min. Hearts are then frozen, powdered, and homogenized into RIPA. Western blotting subsequently resolves the PEGylated protein fraction, that is the putative cell surface fraction, as the fraction of protein shifted to a higher molecular weight. To test whether the NHS reagent significantly crossed the surface membrane (i.e., whether the sarcolemma became significantly permeable during the procedure), we routinely blotted actin, and PEGylation of actin was not detectable in the hearts employed in the results presented in **Figures 2 and 7**.

## Mice

PLM-deficient mice were bred from knockout mice provided by Amy L Tucker (**Jia et al., 2005**) (U Virginia, Charlottesville). DHHC5-GT and DHHC-WT cardiac myocytes were isolated from littermates of heterozygous crosses at F2 or littermates of F2 × F2 crosses of homozygous WT or DHHC5-GT mice (**Li et al., 2011**).

## Analysis of FITC-dextran uptake in intact heart during reoxygenation for 25 min after a 30 min period of anoxia

Confocal imaging of hearts was with a TE2000-U Nikon microscope (60 × oil immersion, 1.45-NA objective; RC-26 recording chamber; Warner Instruments; 40-mW 163-CO2 laser, S Newport

Corporation) at 488 nm using 3% power. Full-frame live recording resolution was 512 × 512 with exposure <1 s (pinhole, 150 μm). Higher resolution captures were 1024 × 1024 with exposures <4 s. Bleaching of fluorophores was negligible. The aorta of isolated hearts was cannulated and retrograde perfusion was implemented with standard oxygenated NIC bath solution at 37°C at a rate of 1.5 ml/min with 50 μM albumin in FA-free experiments or 50 μM albumin plus 100 μM myristate for experiments containing FA. Experiments with 0.1 μM STS or 5 μM CsA contained FA, albumin, and drug through-out. After 20 min control perfusion, anoxic solution was perfused 30 min, and then oxygenated solution was perfused with 100 mg/5 ml of 4000 MW FITC-dextran (Sigma) with flow reduced to 0.5 ml/min. After 25 min, dye-free oxygenated solution was perfused at 1.5 ml/min for 5 min. Then, hearts were transferred to the microscope and perfused at 23°C. During imaging the heart was repositioned several times to gather images from a wide distribution of myocytes. Averages of at least 16 individual myocytes were calculated per frame and at least 10 regions were observed per heart, yielding >160 observations. Fluorescent punctae were counted within each myocyte, and the occurrence of 20 punctae per cell was taken as threshold for the occurrence of MEND.

## Acknowledgements

We thank Amy L Tucker (University of Virginia, Charlottesville) for PLM-deficient mice, Tzu-Ming Wang (UC Berkeley) for software development, and Fang-Min Lu (UT Southwestern, Dallas) for assistance.

## Additional information

### Funding

| Funder | Grant reference number | Author |
| --- | --- | --- |
| National Institutes of Health | HL067942 | Donald W Hilgemann |
| National Institutes of Health | HL513225 | Donald W Hilgemann |

The funders had no role in study design, data collection and interpretation, or the decision to submit the work for publication.

### Author contributions

M-JL, DWH, Conception and design, Acquisition of data, Analysis and interpretation of data, Drafting or revising the article, Contributed unpublished essential data or reagents; MF, Conception and design, Acquisition of data, Analysis and interpretation of data, Drafting or revising the article; J-YL, SLH, Provided DHHC5-GT mice and DHHC5 reagents; GF, Conception and design, Designed and built the NIC-recording device employed

### Ethics

Animal experimentation: This study was performed in strict accordance with the recommendations in the Guide for the Care and Use of Laboratory Animals of the National Institutes of Health. All of the animals were handled according to approved institutional animal care and use committee (IACUC) of UT Southwestern Medical Center at Dallas. The UT Southwestern Medical Center Animal Care and Use Committee approved all animal studies as APN #0376-06-05-1, last renewed on 10/9/12 and expiring on 9/13/15.

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
