## [Decision Letter]

Thank you for sending your work entitled “Massive palmitoylation-dependent endocytosis during reoxygenation of anoxic cardiac muscle” for consideration at *eLife*. Your article has been favorably evaluated by a Senior editor and 3 reviewers, one of whom is a member of our Board of Reviewing Editors.

The following individuals responsible for the peer review of your submission have agreed to reveal their identity: Richard Aldrich (Reviewing editor) and Bertil Hille (peer reviewer).

The Reviewing editor and the other reviewers discussed their comments before we reached this decision, and the Reviewing editor has assembled the following comments to help you prepare a revised submission.

Reoxygenation after anoxia in cardiac tissue results in massive endocytosis (MEND) described in the companion manuscript. Here, the authors extend their MEND experiments to induced cardiomyocytes (iCell cardiomyocytes) in culture during reperfusion injury. Using protocols that either raise cytoplasmic [acyl CoA], increase K succinate, or raise Ca, the authors infer that PTP openings, PKC activation, and a fast reacting calcium-dependent mechanism induce MEND.

MEND is strongly inhibited in mice lacking the acyltransferase, corresponding to significantly preserved right ventricular contractile function in DHHC5 KO mice after reperfusion injury. Finally, phospholemman and flotillin-2 are more palmitoylated after reperfusion injury. The hypothesis is that palmitoylation either reduces Na/K pump activity directly or increases uptake of this transporter.

As in the companion manuscript, the authors study the whole MEND process. In this paper, however, the authors present the interesting finding that reperfusion injury induces MEND, and inhibition of MEND reduces loss of cardiac contractility (at least in R. ventricle). This most consequential finding of a reduction of MEND and preservation of contractility in DHHC5 KO mice suggests a potential new approach to ameliorate the vexing reperfusion injury problem.

The following issues would have to be adequately addressed for the paper to be accepted.

The high importance of the paper is that it brings to real tissue the new hypothesis proposed in the first paper only on the basis of cultured cells, and then it uses that new concept to suggest an approach to relieving reperfusion injury. Several of the mechanistic conclusions are much less well supported and detract from the central point. These aspects of the paper should be diminished to what is strongly supported or should be eliminated.

1) There are two weak parts of the manuscript: 1) the link between palmitoylation and Na/K pump activity and 2) PKC's part in this process. Shotgun palmitoylation of diverse protein targets seems to be the central mechanism, perhaps in analogy to uncontrolled ubiquitination. Somehow, this activates massive membrane uptake, and of course, Na/K ATPase as an abundant membrane protein is lost. The authors need to tone down the centrality of the Na/K ATPase. Although it is logical that this has a major negative impact on cells, it may be simply one of many.

2) Overall, the conclusion that is most interesting is the protection from ischemia reperfusion injury by DHHC5 loss. This makes a good conclusion for the functional importance of palmitoylation, but the authors should remain conservative about what palmitoylation does to loss of protein or membrane.

The only way we see forward is comparison of proteins and their palmitoylation states by mass spec before and after MEND induction in order to obtain more clues about the uptake process. This is likely to take a year or more and should be the topic of follow-up studies.

3) The insufficiently resolved conclusions/speculations on lipid order should be eliminated.

Also increased is IDTPP, the magnitude of a “hydrophobic cation current” per surface area. IDTTP is not explained and no recordings are shown, but perhaps IDTPP is a capacitative displacement current accompanying voltage steps, and it appears when a hydrophobic cation is added to cells and partitions into their plasma membrane. The authors assume that IDTTP assays “disorder” in membranes but don't explain what disorder means in this case, nor why disorder would increase the partition of hydrophobic ion into a plasma membrane. The increase of IDTPP in the hypomorph mice is taken to suggest that “membrane protein palmitoylation promotes membrane order, similar to cholesterol”. It is not clear which protein(s) was/were palmitoylated, nor how such palmitoylation would increase order.

4) In Figure 1, the authors measure the “Na/K pump current”. How this is done is not explained, and apparently not referenced.

It is shown that expression of GFP diminished the pump current by 30%, nearly as much as overexpression of DHHC5 (55%). The effect of GFP is not discussed. Happily, the pump current is doubled by knockdown of both DHHC5 and DHHC2 (the graph seems mislabeled). It is not discussed what the pump current assays in this case. Is the pump current larger because the cytoplasmic Na concentration is increased? Or does lack of DHCC5 increase the surface density of Na pumps, or the fraction of surface pump molecules that are palmitoylated, or all of the above? While Figure 1 is suggestive of a role of palmitoylation, it's difficult to distil a clear message.

5) Figure 2 studies single cardiac cells, and strips of cardiac tissue, from mice with diminished DHHC5 expression. The volume of the myocytes is slightly smaller, and their capacitance per volume slightly larger; what this means is not explained. A capacitance is determined in strips of tissue by “non-invasive capacitance recording” (NIC). That capacitance, too, is increased. NIC is a new method briefly explained in the Methods, but not well enough for others to use it, and it is not validated. It's not explained how a surface area (per volume of an individual cell) is related to what we take to be the summed surface area (not per volume) of all myocytes in a strip of muscle.

6) As with the companion manuscript, the writing, proofreading, and presentation need to be greatly improved.

---

## [Author Response]

*The high importance of the paper is that it brings to real tissue the new hypothesis proposed in the first paper only on the basis of cultured cells, and then it uses that new concept to suggest an approach to relieving reperfusion injury. Several of the mechanistic conclusions are much less well supported and detract from the central point. These aspects of the paper should be diminished to what is strongly supported or should be eliminated*.

I have tried to streamline this article to the issue of reperfusion injury and the relationship of MEND to reperfusion injury.

*1) There are two weak parts of the manuscript: 1) the link between palmitoylation and Na/K pump activity and 2) PKC’s part in this process. Shotgun palmitoylation of diverse protein targets seems to be the central mechanism, perhaps in analogy to uncontrolled ubiquitination. Somehow, this activates massive membrane uptake, and of course, Na/K ATPase as an abundant membrane protein is lost. The authors need to tone down the centrality of the Na/K ATPase. Although it is logical that this has a major negative impact on cells, it may be simply one of many*.

Regarding Na/K pump and palmitoylation, new data now make a more clear picture. In the first figure of the cardiac article, I presented an inverse dependence of Na/K pump currents on DHHC expression in cultured myocytes. It was noted by the reviewers that the inhibition by DHHC5 overexpression was unimpressive because overexpression of GFP caused an inhibition of nearly one-half the magnitude. *I shot myself unnecessarily in the foot!* From many years of experience, we know with great certainty that smaller cells are preferentially transfected when DNA is transfected with Lipofectamine and related reagents. We have already published in detail for these cardiac iCells that the Na/K pump activity is lower in smaller cells. That is because the smaller cells are less well differentiated (Fine and Hilgemann, Am J Physiol Cell Physiol. 2013;305:C481-91), and that is why pump current densities are lower in the GFP-transfected myocytes. In response, I removed data for the untransfected iCells and therewith removed a completely unnecessary complication.

Regarding PKCs, the sticky point that enabled this critique is that the generation of acyl CoAs (subsequent to PTP openings) is required for MEND to occur, but it is definitively not sufficient. I had worked fairly extensively on this issue prior to submission and decided to present clear (and rather extensive) evidence that PKCs play a role in BHK cells. I don’t see why the reviewers objected to that data, except that it burdens the reader with the truth about a biological system that is not entirely simple. The fact is that the equivalent experiments in cardiac myocytes were less impressive, and I am happy to tone them down. This is easily done because I now present much more impressive data that transient oxidative stress supports the final steps of MEND in myocytes (as well as in BHK cells).

I also point the reader toward the idea that the coalescence of proteins and membranes into domains constitutes the development of membrane phase transitions that on their own may have consequences, e.g.cause membrane defects, in addition to consequences of MEND per se.

*2) Overall, the conclusion that is most interesting is the protection from ischemia reperfusion injury by DHHC5 loss. This makes a good conclusion for the functional importance of palmitoylation, but the authors should remain conservative about what palmitoylation does to loss of protein or membrane*.

Well taken. I make it a point to bring out the potential relevance of these findings to real reperfusion injury in real patients.

*The only way we see forward is comparison of proteins and their palmitoylation states by mass spec before and after MEND induction in order to obtain more clues about the uptake process. This is likely to take a year or more and should be the topic of follow-up studies*.

Well, I still find a lot of interesting experiments to do, but you are right. We are initiating mass spec studies and they will indeed take a long time to move forward. The biggest wait will be the wait to be funded to do this seriously! One year, maybe two, if we are lucky, based on my experiences recently.

*3) The insufficiently resolved conclusions/speculations on lipid order should be eliminated*.

*Also increased is IDTPP, the magnitude of a “hydrophobic cation current” per surface area. IDTTP is not explained and no recordings are shown, but perhaps IDTPP is a capacitative displacement current accompanying voltage steps, and it appears when a hydrophobic cation is added to cells and partitions into their plasma membrane. The authors assume that IDTTP assays “disorder” in membranes but don't explain what disorder means in this case, nor why disorder would increase the partition of hydrophobic ion into a plasma membrane. The increase of IDTPP in the hypomorph mice is taken to suggest that “membrane protein palmitoylation promotes membrane order, similar to cholesterol”. It is not clear which protein(s) was/were palmitoylated, nor how such palmitoylation would increase order*.

Agreed. It’s gone! Although it was comforting to me that I could demonstrate changes of a global physical property of the membrane in response to deleting DHHC5.

*In*
Figure 1*, the authors measure the “Na/K pump current”. How this is done is not explained, and apparently not referenced*.

I have referred the reader to the study that was cited in the previous sentence. Measurements were carried out in an identical manner.

*It is shown that expression of GFP diminished the pump current by 30%, nearly as much as overexpression of DHHC5 (55%). The effect of GFP is not discussed. Happily, the pump current is doubled by knockdown of both DHHC5 and DHHC2 (the graph seems mislabeled). It is not discussed what the pump current assays in this case. Is the pump current larger because the cytoplasmic Na concentration is increased? Or does lack of DHCC5 increase the surface density of Na pumps, or the fraction of surface pump molecules that are palmitoylated, or all of the above? While*
Figure 1
*is suggestive of a role of palmitoylation, it's difficult to distil a clear message*.

Regarding Na/K pump and palmitoylation, new data now make a clearer picture. In the first figure of the cardiac article, I presented an inverse dependence of Na/K pump currents on DHHC expression in cultured myocytes. It was noted by the reviewers that the inhibition by DHHC5 overexpression was unimpressive because overexpression of GFP caused an inhibition of nearly one-half the magnitude. From many years of experience, we know with great certainty that smaller cells are preferentially transfected when DNA is transfected with Lipofectamine and related reagents. We have already published in detail for these cardiac iCells that the Na/K pump activity is lower in smaller cells. That is because the smaller cells are less well differentiated (Fine and Hilgemann, Am J Physiol Cell Physiol. 2013;305:C481-91), and that is why pump current densities are lower in the GFP-transfected myocytes. In response, I removed data for the untransfected iCells and therewith removed a completely unnecessary complication.

Next, I have worked more extensively with the DHHC5-deficient mice and have been able to simplify Figure 2 of the cardiac article with no compromise to the science. Importantly, I had not noticed before that DHHC5 mice grow significantly more slowly than their WT littermates. Actually, Dr. Hofmann had a complete file of data on this point, and I have reported numbers on the growth defect in the revised article. Accordingly, the myocytes from DHHC5-GT mice were somewhat smaller than WT control myocytes, specifically because we routinely use mice at about 2 months of age to avoid escalating housing costs. By simply waiting two weeks longer, the myocytes are of the same size, capacitance is then significantly increased in the DHHC5-GT myocytes, pump current densities are increased, and the localization of phospholemman (PLM, and therefore Na/K pumps) to the surface membrane is increased by nearly the same percentage as the pump current densities. Given that myocyte area is rather marginally increased, the results together make a strong case that pump localization is a major factor in the change of current density. Clearly, this dataset begs us to analyze other conductances and E/C coupling function, but this is not reasonable in the context of the present articles.

*5)*
Figure 2
*studies single cardiac cells, and strips of cardiac tissue, from mice with diminished DHHC5 expression. The volume of the myocytes is slightly smaller, and their capacitance per volume slightly larger; what this means is not explained. A capacitance is determined in strips of tissue by “non-invasive capacitance recording” (NIC). That capacitance, too, is increased. NIC is a new method briefly explained in the Methods, but not well enough for others to use it, and it is not validated. It's not explained how a surface area (per volume of an individual cell) is related to what we take to be the summed surface area (not per volume) of all myocytes in a strip of muscle*.

I have worked more extensively with the DHHC5-deficient mice and have been able to simplify Figure 2 of the cardiac article with no compromise to the science. Importantly, I had not noticed before that DHHC5 mice grow significantly more slowly than their WT liter mates. Actually, Dr. Hofmann had a complete file of data on this point, and I have reported numbers on the growth defect in the revised article. Accordingly, the myocytes from DHHC5-GT mice were somewhat smaller than WT control myocytes, specifically because we routinely use mice at about 2 months of age to avoid escalating housing costs. By simply waiting two weeks longer, the myocytes are of the same size, capacitance is then significantly increased in the DHHC5-GT myocytes, pump current densities are increased, and the localization of phospholemman (PLM, and therefore Na/K pumps) to the surface membrane is increased by nearly the same percentage as the pump current densities. Given that myocyte area is rather marginally increased, the results together make a strong case that pump localization is a major factor in the change of current density. Clearly, this data set begs us to analyze other conductances and E/C coupling function, but this is not reasonable in the context of the present articles.

Regarding NIC recording, I introduce the details now in a clearer fashion at the beginning of the Results section. I point out immediately that all results employing NIC recording have been confirmed by either patch clamp measurements using myocytes or by optical methods using intact hearts. Furthermore I point the reader to results presented in the Methods that further validate the NIC methods. I explain to the reader that the NIC method was critical because it allowed me to analyze the behavior of intact cardiac tissue under wide range of conditions. The method was key to moving forward with this paper, even though everything has been confirmed with other methods. I believe that the presentation of the data from NIC recording is very well justified and that the description of the method and potential problems in using are adequately described for this publication.

*6) As with the companion manuscript, the writing, proofreading, and presentation need to be greatly improved*.

I have really tried hard to improve the clarity of this article. The overwhelming recommendation of the reviewers for both articles was to work hard on the presentation, to improve the clarity of writing and figures, and to make the articles more attractive to a wider audience. I have really tried:

I have remade all of the figures, using color and cartoons to make them both easier to understand and to make them friendlier to look at on a computer screen. I was at first skeptical, but my colleagues strongly agree that the improvements are big ones.

I completely rewrote both the Introduction and the Discussion of this paper. As noted already, I have taken your recommendation to get off the Na/K horse seriously and I launch the paper straight from the perspective of ‘reperfusion injury.’ I also try to guide the discussion along this line, which likely will attract the most readers. As for Na/K pumps, I do keep the perspective in foreground that this transporter is a key regulator of cardiac contractility and therefore that it deserves general attention. That’s a fact.

In both articles I have included many didactic lines to help guide the general reader through the data (and to understand what we do experimentally) both in the text of the article and in the figure legends.